# Apparent bias toward long gene misregulation in MeCP2 syndromes disappears after controlling for baseline variations

Ayush T. Raman[1,2], Amy E. Pohodich [2,3], Ying-Wooi Wan[2,4], Hari Krishna Yalamanchili[2,4], William E. Lowry[5], Huda Y. Zoghbi [2,3,4,6] & Zhandong Liu [1,2,7]

Recent studies have suggested that genes longer than 100 kb are more likely to be misregulated in neurological diseases associated with synaptic dysfunction, such as autism and Rett syndrome. These length-dependent transcriptional changes are modest in MeCP2-mutant samples, but, given the low sensitivity of high-throughput transcriptome profiling technology, here we re-evaluate the statistical significance of these results. We find that the apparent length-dependent trends previously observed in MeCP2 microarray and RNA-sequencing datasets disappear after estimating baseline variability from randomized control samples. This is particularly true for genes with low fold changes. We find no bias with NanoString technology, so this long gene bias seems to be particular to polymerase chain reaction amplification-based platforms. In contrast, authentic long gene effects, such as those caused by topoisomerase inhibition, can be detected even after adjustment for baseline variability. We conclude that accurate characterization of length-dependent (or other) trends requires establishing a baseline from randomized control samples.

[1] Graduate Program in Quantitative and Computational Biosciences, Baylor College of Medicine, Houston, TX 77030, USA. [2] Jan and Dan Duncan Neurological Research Institute at Texas Children's Hospital, Houston, TX 77030, USA. [3] Department of Neuroscience, Baylor College of Medicine, Houston, TX 77030, USA. [4] Department of Molecular and Human Genetics, Baylor College of Medicine, Houston, TX 77030, USA. [5] Department of Molecular, Cell and Developmental Biology, University of California, Los Angeles, Los Angeles, CA 90095, USA. [6] Howard Hughes Medical Institute, Baylor College of Medicine, Houston, TX 77030, USA. [7] Department of Pediatrics, Section of Neurology, Baylor College of Medicine, Houston, TX, USA. These authors contributed equally: Ayush T. Raman, Amy E. Pohodich. Correspondence and requests for materials should be addressed to H.Y.Z. (email: hzoghbi@bcm.edu) or to Z.L. (email: zhandong.liu@bcm.edu)

L arge-scale analysis of transcriptional changes has transformed our understanding of many human neurological diseases. Neurodevelopmental disorders such as Rett syndrome (RTT) and Fragile X syndrome, for example, involve transcriptional alterations in thousands of genes[1]. This is not surprising in the case of RTT, given the role of the causative gene, MeCP2, in epigenetic regulation. But recent microarray and RNA-sequencing (RNA-seq) studies have observed a trend that is surprising: the genes dysregulated in neurodevelopmental syndromes tend to be those that are longer than 100 kb[2,3]. This intriguing length bias has been observed across both epigenetic and transcriptional datasets for Angelman syndrome[4], RTT[5–8], Fragile X syndrome[9], and autism[10,11]. The degree of bias tends to be fairly mild, however, and long genes are themselves over-represented in the brain compared to other tissues in the body[2]. Because this is a recurring theme in neurologic disease datasets, it is worth examining this apparent bias more closely.

The aforementioned gene expression studies[5,6,10,11] partitioned the entire genome into hundreds of overlapping bins (or windows), with each bin containing hundreds of genes. Within each bin, the average fold change in wild-type (WT) or untreated brain tissue was compared to that observed in the knock-out or treatment groups, and a running average log$_2$fold change was plotted against the average gene length. In these running average plots, long genes demonstrated a nonzero mean compared to short genes. Yet these analyses did not establish a baseline of inherent variation among samples within a given genotype, and they did not employ a statistical test to determine the significance of the length-dependent changes. Variations in measured gene expression can arise because of RNA priming[12,13], guanine–cytosine content[14], transcript length[15], or library preparation[16], all of which must be accounted for before drawing biological conclusions[17,18].

We, therefore, reanalyze a number of large datasets derived from different transcriptome profiling technologies and set out to determine the best way to enhance the signal-to-noise ratio. To this end, we develop a statistical approach to accurately estimate noise and identify statistically significant gene length-dependent changes. Upon implementing this approach, we show a genuine trend in transcriptional alterations in long genes when the fold-change values are large, such as those caused by topoisomerase inhibition. In contrast with prior studies, however, we find no preferential misregulation of long genes in MeCP2 datasets after correcting for statistical significance and baseline variability. We propose that smaller fold changes in transcription observed after polymerase chain reaction (PCR) amplification leads to over-estimation of long gene expression levels.

## Results

**Baseline length dependency should be estimated from controls.** Preferential dysregulation of long genes has generally been estimated by computing the average gene expression fold changes between experimental groups and plotting this fold change against the gene length[5,6,10], also known as running average plots (red curve in Fig. 1a). It is worth noting that the statistical significance of running average plots has never been evaluated in the current literature. We decided to estimate statistical significance by constructing a null distribution of the running average plot from randomized control samples (Supplementary Fig. 1).

We first analyzed data from a study that evaluated the transcriptional effects of the topoisomerase I inhibitor topotecan in autism[10]. When we constructed a running average plot comparing the gene expression changes between topotecan drug-treated neurons (drug or D) and vehicle-treated cortical neurons (vehicle or V), we observed a preferential downregulation of long genes (the running average plot comparing drug vs. vehicle is indicated by the red curve in Fig. 1a; Supplementary Fig. 1). To estimate the baseline variation among control samples, we compared two random sets of vehicle-treated cultured cortical neurons to each other (blue curve in Fig. 1a). Given that these untreated samples were obtained from littermates, we did not expect to observe any differences in gene expression and predicted that a running average plot comparing gene expression between vehicle-treated control samples would yield a horizontal line through $y = 0$. We found, however, that genes over 100 kb in length tended to be downregulated (blue curve in Fig. 1a). This effect was found for both RNA-seq and microarray datasets (Fig. 1a; see Table 1 for details about comparisons made in Fig. 1) and indicates that a portion of the length-dependent trend observed in the topotecan datasets is due to a length-dependent bias (i.e., noise) that can be observed even in the control samples.

To determine the significance of average fold-change trends, we applied a Student's $t$ test to each of the matching data bins from the drug vs. vehicle (D/V) and vehicle vs. vehicle (V/V) comparisons, followed by an adjustment for multiple hypothesis testing. For consistency, these plots are referred to as overlap plots (Supplementary Fig. 1, Methods). At a false discovery rate (FDR) of 0.05, only the long gene bins in both RNA-seq and microarray datasets showed statistically significant preferential downregulation following topotecan treatment (lower half of charts in Fig. 1a, red dots indicate statistically significant bins; Supplementary Fig. 1). Therefore, although the control samples showed that long genes are downregulated at baseline (i.e., when comparing controls to controls), topotecan treatment produced an even stronger downregulation of long genes, providing sufficient signal to overcome the noise (or intrasample variation) observed in long genes at baseline. These datasets[10,19] established a statistical procedure and provided positive controls for further analyses of long gene trends in other studies.

**Long gene trends disappear in MeCP2 mouse datasets.** Studies of MeCP2-related disorders—both Rett syndrome (caused by loss-of-function mutations in *MECP2*) and *MECP2* duplication syndrome (caused by duplication or triplication of the locus)—have provided a wealth of transcriptome data. Experiments in mouse models of both syndromes have suggested that loss of MeCP2 function causes preferential upregulation of long genes[5,6] and, conversely, that gain of MeCP2 function leads to preferential downregulation of long genes[6]. We chose to delve deeper into these datasets to explore the extent of the contribution of long genes to RTT pathology. We first applied our approach to ten publicly available MeCP2 studies (see Supplementary Data 1; in total 15 gene expression datasets) across 28 different tissue types[5,6,8,20–26]. We first computed the running average plots and were able to reproduce the same results as reported previously[5,6]. However, the baseline variation between WT samples (blue curves in Fig. 1b) extensively overlaps with the running average plots from the *Mecp2*-null (KO) samples (red curves in Fig. 1b; see also Supplementary Figs. 2a–k). This overlap indicates that the signal originally reported for the KO vs. WT comparison can be largely explained by noise (or intra-sample variation) in the dataset, as there is no clear separation between the WT vs. WT curves and the KO vs. WT curves in most brain regions surveyed.

A dozen long gene bins showed significant preferential upregulation in *Mecp2*-null mice in these datasets (FDR < 0.05; Fig. 1b, right panel)—but so did an even larger number of bins for genes less than 100 kb in length (Fig. 1b, c; see also Supplementary Figs. 2b, c and 2f, g). No preferential repression of long genes was observed in datasets from *MECP2*-over-expression models (Tg model with human *MECP2*) (Fig. 1c; see

also Supplementary Fig. 2l)—on the contrary, we found more short genes to be preferentially dysregulated in the *MECP2*-overexpression models (Fig. 1c). Thus, when assessing the bins of genes with a significant difference in expression between WT mice and mice with abnormal MeCP2 levels, we found that genes with a variety of lengths were altered in KO and Tg mice. While there are certainly some long genes with significantly altered

expression in both KO and Tg mice, there is no consistent and preferential long gene trend in the MeCP2 datasets.

**Nuclear RNA profiles of RTT mice models lack long gene trend**. A recent study reported that transcripts of long genes were downregulated in nuclear and nascent RNA samples[8] from the cortical cells of *Mecp2*-mutant mice bearing either of two Rett-

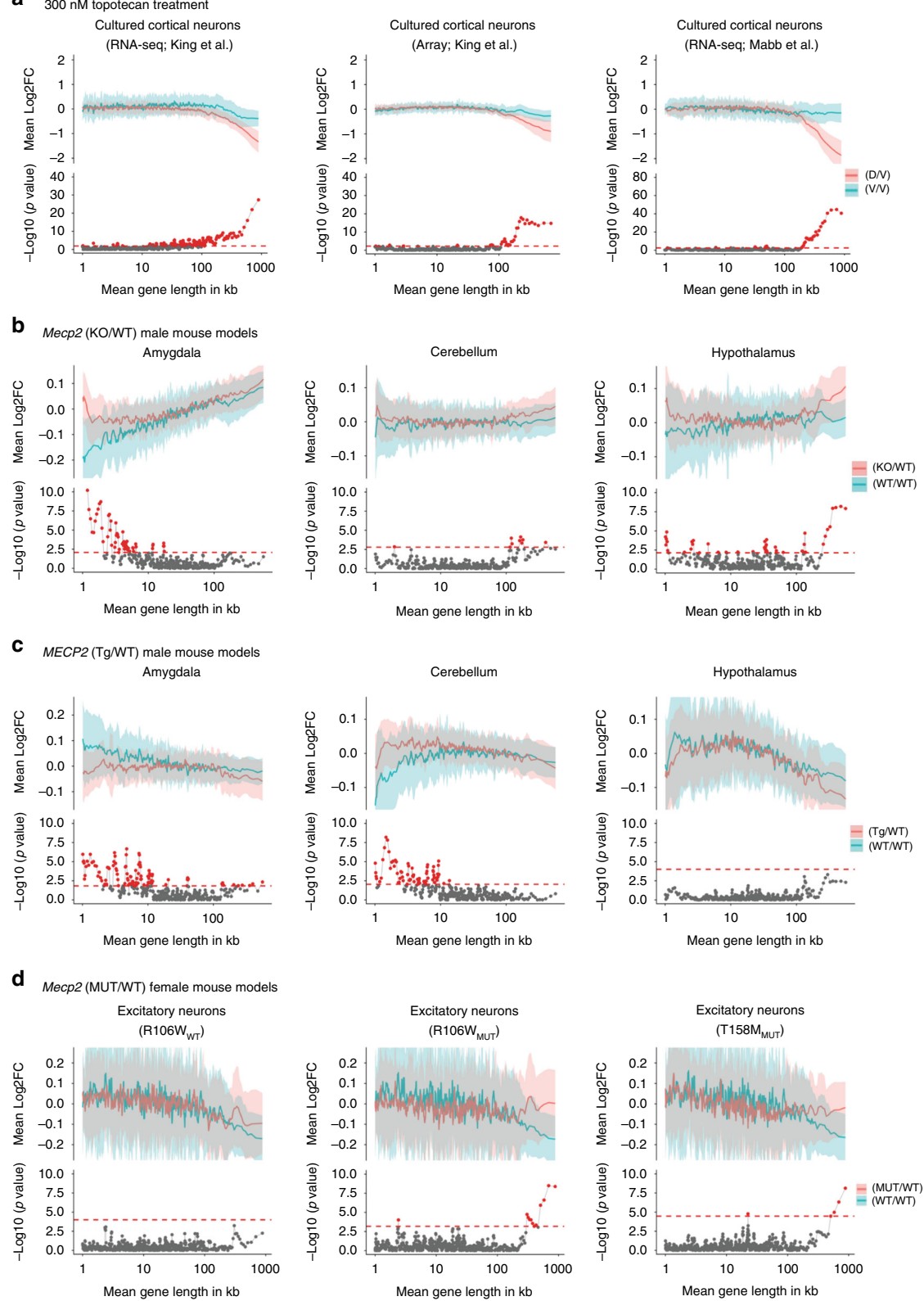

**Table 1 List of comparisons used in overlap or average plots in Figs. 1 and 2**

| Brain region | Mouse strain/human samples compared | Reference |
|---|---|---|
| Fig. 1a. Cultured cortical neurons (left panel) | BL: hybrid vehicle vs. hybrid vehicle ($n = 2$ each)<br>RL: hybrid topotecan vs. hybrid vehicle ($n = 5$ each) | 10 |
| Cultured cortical neurons (middle panel) | BL: hybrid vehicle vs. hybrid vehicle ($n = 1$ each)<br>RL: hybrid topotecan vs. hybrid vehicle ($n = 3$ each) | 10 |
| Cultured cortical neurons (right panel) | BL: hybrid vehicle vs. hybrid vehicle ($n = 1$ each)<br>RL: hybrid topotecan vs. hybrid vehicle ($n = 3$ each) | 19 |
| Fig. 1b. Amygdala (left panel) | BL: C57BL WT vs. C57BL WT ($n = 2$ each)<br>RL: C57BL KO vs. C57BL WT ($n = 5$ each) | 24 |
| Cerebellum (middle panel) | BL: C57BL WT vs. C57BL/6J WT ($n = 2$ each)<br>RL: C57BL KO vs. C57BL/6J WT ($n = 5$ each) | 21 |
| Hypothalamus (right panel) | BL: C57BL WT vs. C57BL/6J WT ($n = 2$ each)<br>RL: C57BL KO vs. C57BL/6J WT ($n = 4$ each) | 22 |
| Fig. 1c. Amygdala (left panel) | BL: FVB WT vs. FVB WT ($n = 2$ each)<br>RL: FVB KO vs. FVB WT ($n = 5$ each) | 24 |
| Cerebellum (middle panel) | BL: FVB WT vs. FVB WT ($n = 2$ each)<br>RL: FVB KO vs. FVB WT ($n = 5$ each) | 21 |
| Hypothalamus (right panel) | BL: FVB WT vs. FVB WT ($n = 2$ each)<br>RL: FVB KO vs. FVB WT ($n = 4$ each) | 22 |
| Fig. 1d. Cortical excitatory neurons R106W$_{WT}$ (left panel) | BL: C57BL WT vs. C57BL WT ($n = 1$ each)<br>RL: C57BL R106W$_{WT}$ vs. C57BL WT ($n = 2$ each) | 8 |
| Cortical excitatory neurons R106W$_{MUT}$ (middle panel) | BL: C57BL WT vs. C57BL WT ($n = 1$ each)<br>RL: C57BL R106W$_{MUT}$ vs. C57BL WT ($n = 2$ each) | 8 |
| Cortical excitatory neurons T158M$_{MUT}$ (right panel) | BL: C57BL WT vs. C57BL WT ($n = 1$ each)<br>RL: C57BL T158M$_{MUT}$ vs. C57BL WT ($n = 2$ each) | 8 |
| Fig. 2a. iPSC (left panel) | BL: iPSC WT vs. iPSC WT ($n = 2$ each)<br>RL: iPSC RTT ($n = 4$) vs. iPSC WT ($n = 5$ each) | GSE107399 |
| NPC (middle panel) | BL: NPC WT vs. NPC WT ($n = 2$ each)<br>RL: NPC RTT ($n = 4$) vs. NPC WT ($n = 5$ each) | GSE107399 |
| Neuron (right panel) | BL: neuron WT vs. neuron WT ($n = 2$ each)<br>RL: neuron RTT ($n = 4$ each) vs. neuron WT | GSE107399 |
| Fig. 2b. Frontal cortex (left panel) | RL: postmortem RTT vs. controls ($n = 3$ each)<br>BL: postmortem pooled sample from 2- and 4-year-old patient vs. control ($n = 1$ each) | 29 |
| Frontal cortex (right panel) | GL: postmortem pooled sample from 5-year-old patient vs. age-matched control ($n = 1$ each)<br>PL: postmortem pooled sample from 8-year-old patient vs. age-matched control ($n = 1$ each) | 29 |
| Fig. 2c. Frontal cortex (left panel) | RTT female samples compared to age-matched controls (ages 17–20 years; $n = 3$) | 31 |
| Temporal cortex (right panel) | RTT female samples compared to age-matched controls (ages 17–20 years; $n = 3$) | 31 |
| Fig. 2d. Frontal cortex (left panel) | RTT female samples compared to age-matched controls (ages 18 years; $n = 1$ each) | GSE107399 |
| Frontal cortex (right panel) | RTT male samples (age 1 year) to compared to age-matched (age 2 day) controls ($n = 1$ each) | GSE107399 |

Hybrid refers to C57BL/6J × CASTEi/J F1-hybrid mice. BL, RL, GL, and PL stand for blue line, red line, green line, and purple line, respectively

**Fig. 1** Establishing baseline length-dependent trends and comparison of MeCP2 microarray and RNA-seq datasets. **a** Topotecan datasets[10,19]. The top half of each subgraph: comparison of gene expression between cultured cortical neurons from C57BL/6J × CASTEi/J F1-hybrid mice that were treated with vehicle (V) or with other vehicle-treated samples (V/V, blue line), and comparison of topotecan-treated cortical neurons (D) with vehicle-treated samples (D/V, red line). **b**–**d** Mecp2 datasets. Note the change in the scale of the y-axis. **b** Mecp2-KO datasets. Top half of subgraphs: comparison of gene expression between two sets of WT male C57BL samples (blue line) and comparison of gene expression between male Mecp2-null (KO) and WT male littermates (red line) in amygdala[24], cerebellum[21], and hypothalamus[22]. **c** MECP2-overexpression (Tg) datasets. The top half of each subgraph: comparison of gene expression between two sets of WT male FVB samples (blue line) and comparison of expression between Tg samples and WT littermates (red line) in amygdala[24], cerebellum[21], and hypothalamus[22]. **d** Cortical excitatory neurons from Mecp2-heterozygous female mice. The top half of each subgraph: comparison between two sets of WT C57BL samples (blue line), and the comparison between WT samples and WT neurons from R106W-heterozygous mice (left panel), between WT neurons and Mecp2-mutant neurons with the R106W mutation (middle panel), and between WT neurons and Mecp2-mutant neurons bearing the T158M mutation (right panel)[8]. The blue or red line represents fold-change in expression for genes binned according to gene length (bin size: 200 genes; shift size: 40 genes[6]). The blue and red shaded areas correspond to one-half of one standard deviation of each bin and the bottom half of each subgraph is the P value from two-sample t test between D/V and V/V, KO/WT and WT/WT, Tg/WT and WT/WT, and MUT/WT and WT/WT in (A–D), respectively. Red dots: bins with FDR < 0.05. The red dashed line at the bottom of subgraphs indicates the minimum $-\log_{10}(P$ value) that corresponds to a FDR (false discovery rate) < 0.05. See Table 1 for comparison details and sample numbers. See Supplementary Fig. 1 for our statistical approach and Supplementary Figs. 2–4 for the additional analyses from published MeCP2 datasets

causing mutations, T158M or R106W, which are among the most common mutations found in RTT patients[27]. The dataset was generated by combining an in vivo biotinylation system with Cre-loxP technology that circumvented the cellular heterogeneity of the brain and enabled examination of transcriptomic changes due to MeCP2 in specific cell types, in both male and female mice[8].

We reanalyzed the data using overlap plots and observed no significant downregulation of long genes in WT or *Mecp2*-mutant excitatory neurons from 18-week-old T158M or R106W female mice (Fig. 1d; see also Supplementary Fig. 3e). In excitatory neurons bearing the R106W mutation, we observed a few bins that differed significantly from WT expression levels. These bins with significant gene expression changes were not due to the downregulation of long genes in mutant samples, however; rather, these bins were significant due to the downregulation of long genes in control (WT) samples, as indicated by the downward slopes of the running average plots comparing WT vs. WT samples (blue lines in Fig. 1d and Supplementary Fig. 3e). Similarly, we observed no significant repression of long genes in nuclear RNA-seq datasets from excitatory and inhibitory neurons from 6-week old male mice with the same mutation type (Supplementary Figs. 3a–d). Finally, when we examined down-regulation of long genes from the GRO-seq (global nuclear run-on with high-throughput sequencing) data collected from these mice, we confirmed a marginal significance in the downregulation of long genes, but upregulation of long genes was not observed in whole-cell RNA-seq data (Supplementary Fig. 4).

Together, these results suggest that when the fold-change difference is 50% or more, as it is in the topotecan datasets, there is likely to be a genuine long gene bias. But when the fold-change effect is small (<15%), as it is with the long genes observed in the MeCP2 datasets, it is more likely that the observed long gene trend is due to inherent variation among samples. The reported long gene trend in the MeCP2 datasets is in the same range as the noise in the intra-sample comparison in the control groups, and this effect was seen in all the MeCP2 datasets that we assessed. This further suggests that the length-dependent variability estimated from microarray and RNA-seq platforms is not sensitive enough to capture small transcriptional changes. We therefore recommend that baseline gene-length dependency be evaluated from the control group first to understand the statistical significance of observed long gene trends in any sequencing dataset.

**Possible influence of age in human *MECP2* datasets**. To determine whether preferential dysregulation of long genes occurs in in vitro human Rett datasets, we computed overlap plots on samples from isogenic human induced pluripotent stem cells (iPSCs), neural progenitor cells, and neurons from the fibroblasts of two independent patients, with and without a *MECP2* mutation[28] (Supplementary Data 2). We found no pre-ferential upregulation of long genes in these in vitro samples (Fig. 2a; see Table 1 for details about comparisons made in Fig. 2), but we did see a trend toward downregulation of long genes among human in vitro RTT neuron samples based on running average plot only, which is contrary to reports from *Mecp2*-null mouse models[5,6].

Although long genes do not appear to be upregulated above the level of background noise in murine *Mecp2* datasets, they have been reported to be preferentially upregulated in human RTT samples[6], and we wondered if a more robust signal would be observed in postmortem human datasets. Three RTT and three normal control samples from the superior frontal gyrus were obtained from a previous study[29]. These samples were from three different ages: RTT samples were obtained from donors aged 8, 6,

and <4 years (pooled samples from a 2- and a 4-year old), with approximate age-matched normal control samples obtained from donors aged 10, 5, and 2 years, respectively. The long gene trend was observed in a comparison of the three RTT samples to the three control samples (Fig. 2b). Because the brain changes markedly from ages 1 to 5 years[30] in both RTT patients and healthy children, we reanalyzed the data by comparing each individual sample to its age-matched control. Dysregulation of long genes was observed only in the 2- and 4-year-old RTT samples (Fig. 2b left panel), but not in either the 5- or 8-year-old RTT samples (Fig. 2b right panel). Unfortunately, the statistical significance of this observation cannot be established because of the small sample size ($n = 1$ each).

To determine whether length-dependent misregulation of long genes occurs in other human datasets, we analyzed samples from another study[31] and in-house generated RNA-seq RTT datasets. The Lin et al. dataset[31] consists of postmortem brain samples from the frontal and temporal cortex of RTT patients with age-matched controls (ages 17 and 20 years, $n = 3$ each). Because the phenotypes are similar for RTT patients in this age range[30], we grouped these RTT samples together and compared them to the pooled age-matched controls. We computed running average plots on the normalized dataset (Methods), and observed no overrepresentation of long genes (Fig. 2c). Similar results were reported by the original study[31]. Consistent with our previous results, there was no long gene trend in the running average plot of the RNA-seq RTT dataset collected from a postmortem frontal cortex sample obtained from an 18-year-old RTT female (Fig. 2d, left panel) when it was compared to its age-matched control (age = 18 years, $n = 1$ each). To further probe whether the long gene trend might be present in the early stages of the disease, we compared a RTT postmortem male sample from frontal cortex (age = 1 year, $n = 1$) to an age-matched control sample (age = 2 days, $n = 1$) and again found no significant upregulation of long genes (Fig. 2d, right panel).

One possible explanation for the lack of a long gene trend in human RTT samples is heterogeneity among the various samples (including differences in genetic background), which increases the inherent variability in gene expression among biological replicates. Such variability could obscure the effects of a subtle bias in the sequencing process. Nevertheless, the present findings suggest that long genes are not preferentially misregulated in human RTT datasets.

**Gene expression analyses in Topotecan and MeCP2 datasets**. Our analyses suggest that current transcriptome profiling tech-nologies are limited in their ability to detect subtle differences in gene expression. We hypothesize that long gene effects, if genu-ine, should be apparent in both binning analyses and the tradi-tional differential gene expression analyses. We, therefore, decided to focus our attention on the genes previously reported to be differentially expressed[10,19–23]. We divided the entire list of differentially expressed genes into four groups based on gene length (> or <100 kb) and fold-change direction (either up or down). Consistent with our overlap plots, we found long genes to be substantially overrepresented and downregulated in topotecan datasets (Fig. 3a). This result proves that our approach does detect long gene trends when they actually exist. In the *Mecp2* datasets, however, we did not find a preferential upregulation of long genes (Fig. 3b–d) except in the hippocampal dataset[20] (Supplementary Fig. 5). Another important difference between the topotecan and MeCP2 datasets was that short genes domi-nated among all differentially expressed genes in MeCP2 datasets (Fig. 3b–d; Supplementary Fig. 5). This further supports the notion that a preference for long gene misregulation is not an

inherent feature of gene expression following *Mecp2* disruption. This is not to say that MeCP2 does not regulate a subset of long genes, only that our analysis found no preferential misregulation of long genes in RTT mouse models.

**SEQC benchmark datasets are prone to long gene bias.** To investigate whether the apparent length bias might be a function of amplification-based platforms, we next performed running average analysis on the samples from the phase-III sequencing/microarray quality control (SEQC) project[32]. SEQC was designed to evaluate the performance of various sequencing platforms,

sources of bias in gene expression samples, and various methods for downstream analysis. The consortium generated benchmark datasets using four different types of RNA samples: A (Universal Human Reference RNA), B (Human Brain Reference RNA), C (a mixture of A and B at a ratio of 3:1), and D (a mixture of A and B at a ratio of 1:3). The RNA-seq datasets generated using the Illumina HiSeq 2000 platform across six different sites were used for quality control analyses (Methods), and the raw read counts were normalized using the DESeq2 method[33].

To determine whether the SEQC dataset showed nominal batch effects or other nonbiological variability, we used multi-dimensional scaling (MDS) plots to see if the samples clustered

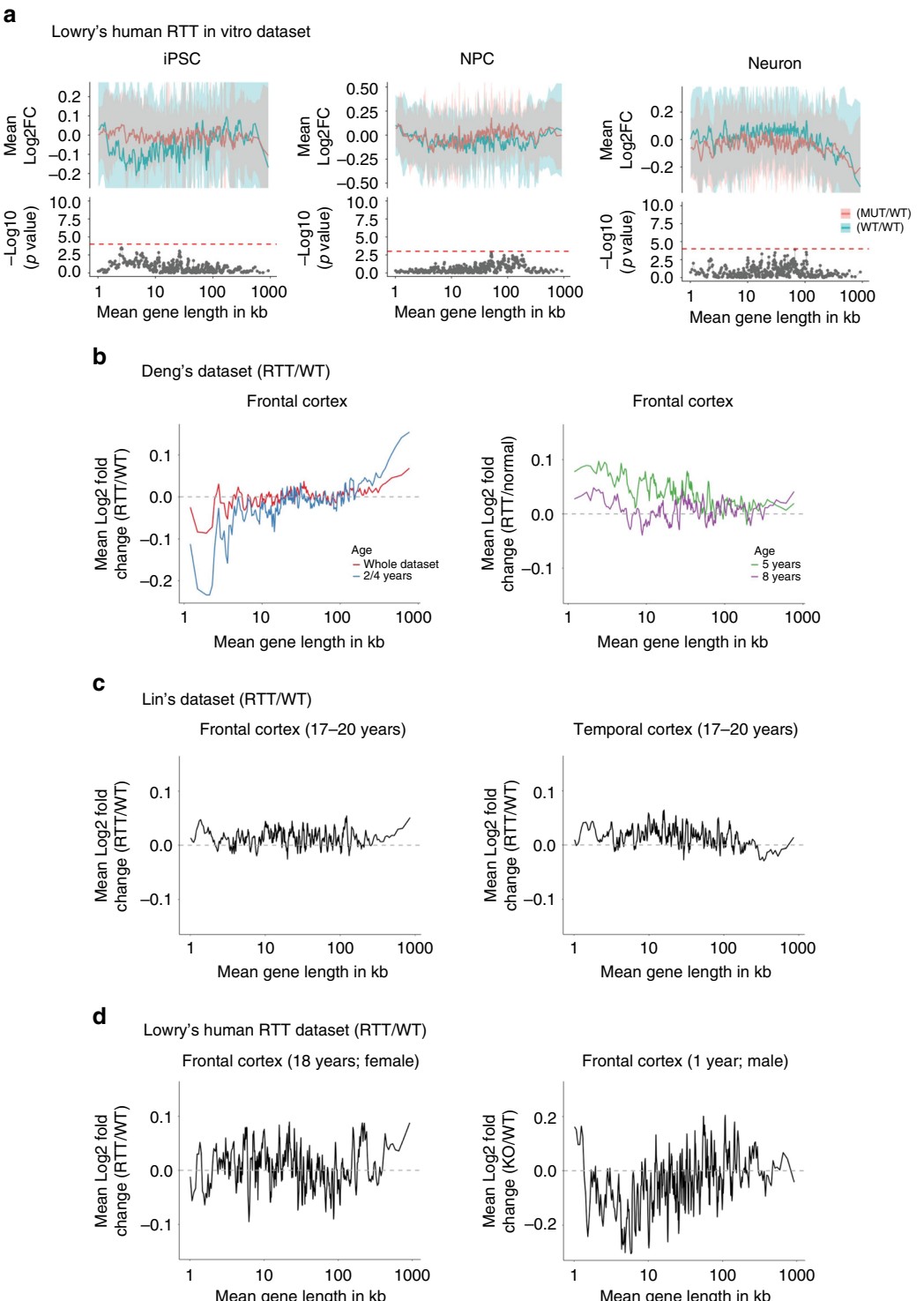

according to RNA sample type (Fig. 4a). To ascertain whether or not the samples were consistently titrated, we calculated the $\beta$ ratio of observed gene expression in the samples, which is obtained from the following equation: $((B–A)/(C–A))$[32]. The value of the $\beta$ ratio[34] is 4:1 (or $\log_2(4) = 2$). In theory, the $\beta$ ratio should be independent of gene length in the brain and nonbrain tissues. After assessing various SEQC datasets, we found that the Novartis dataset had nominal batch effects and the $\beta$ ratio was close to 2. This dataset would be ideal, as it would not bias downstream analyses.

We then investigated whether the $\beta$ ratio computed from all SEQC sample types is constant over gene length. The expected average $\log_2 \beta$ ratio should be a horizontal line along the $x$-axis with a $y$-intercept equal to two (i.e., $y = 2$ on an $xy$ plane). We found, however, that the expected ratio was not maintained for long genes and was overestimated (Fig. 4b). Moreover, we observed a similar bias in the $\beta$ ratio with respect to transcript length, with longer transcripts being overrepresented (Supplementary Fig. 6a, right panel). Overall, the range of overestimation in the RNA-seq dataset was between 3 and 40%. Consistent with our findings, another study (using a different dataset) previously reported that long genes were more likely to be identified as statistically significant in RNA-seq datasets[15].

To determine whether this observed length-dependent bias is due to the fact that long genes are highly expressed in brain tissue, we separated human brain reference (sample type B) RNA-seq samples into two groups of 32 samples each, based on their $y$-axis coordinates on the MDS plot, and computed a running average plot. Since these samples were technical replicates of the same reference RNA sample type, we expected the mean $\log_2$ fold change to be a horizontal line along the $x$-axis with a $y$-intercept equal to zero (i.e., $y = 0$ on an $xy$ plane). Instead, we found that long genes deviated from the expected pattern, with the fold changes of long genes being overestimated (Supplementary Fig. 6a, left panel).

To determine whether the long gene bias was unique to the RNA-seq datasets or could be detected on other platforms, we investigated the MAQC-III microarray Affymetrix dataset generated by the SEQC consortium[32]. Human brain reference samples (type B) were separated into two groups based on y-axis location on the MDS plot (Fig. 4c). The running average plots were computed against their average gene length using the same parameters as described for the RNA-seq analysis above. As with the RNA-seq samples, the average fold change for long genes deviated from the expected value of zero (Supplementary Fig. 6b, left panel). When the $\beta$ ratio was plotted against the mean gene length (Fig. 4d) or mean transcript length (Supplementary Fig. 6b right panel), we found that long genes were overrepresented. We also observed a long gene bias in both RNA-seq and microarray

datasets in a comparison of two groups of Universal Human Reference samples (Supplementary Fig. 6a, b, middle panel). The overestimation in the microarray dataset ranged from 1.5 to 23% —lower overall than for the RNA-seq dataset, but indicating that microarray datasets are also predisposed to gene and transcript length-dependent biases.

**Long gene bias is independent of normalization methods**. To ensure that the long gene bias we observed was not due to our normalization methods, we compared the mean $\log_2$ fold change using three different normalization techniques: total count, DESeq[35], and edgeR/TMM[36,37]. We normalized the raw read counts from four different RNA sample types using each of the three normalization methods and computed running average plots of the $\beta$ ratios against gene and transcript length. In all cases, long genes were still overestimated, regardless of the normalization method (Supplementary Fig. 7a, b). This lends support to the notion that the overrepresentation of long genes is independent of the normalization technique.

**Long gene bias is not observed in NanoString datasets**. We hypothesized that PCR amplification, a process shared by both microarray and RNA-seq technologies, might introduce the observed bias in long gene expression. Therefore, we performed NanoString nCounter gene expression quantification, a technique that does not use amplification, using the SEQC reference RNA samples (A– D) ($n = 6$ each). The MDS plot on normalized data showed that the samples clustered based on sample type (Supplementary Fig. 8a), and the effect of batches was minimal (Methods). The code set consisted of ~184 long genes, out of which ~132 long genes were expressed in brain samples (Supplementary Fig. 8b). We again computed the running average plots against their average gene length, and we observed no long gene bias between the brain samples or when computing the $\beta$ ratio of the samples (Supplementary Fig. 8c, d).

We next compared the mean expression levels of all the common genes across the RNA-seq, microarray and nCounter datasets. Our analysis showed that fold changes of long genes are overestimated in the RNA-seq ($P$ value $< 2.7e−07$; Fig. 4e) and microarray datasets ($P$ value $< 0.021$; Fig. 4f); in contrast, the nCounter dataset showed no difference in the average expression of long and short genes ($P$ value $= 0.86$; Fig. 4g). Although it is possible that the smaller number of genes (~680) might make it more difficult to detect a preference, the proportion of long genes in this dataset (~180 out of ~680 genes, or 26%) is twice that found in the human transcriptome (~3200 long genes out of ~24,000 genes, or 13%). Any preference for long genes should thus be revealed even more strongly in this dataset. These results lead us to posit that the long gene overestimation we observed in

**Fig. 2** No bias toward long genes is detected in *MECP2* human datasets. **a** RNA-seq analysis of isogenic human Rett syndrome in vitro models. Overlap plots were used to compare WT and KO samples, where the top half of each subgraph shows the comparison of gene expression between WT samples and other WT samples (blue line), and the comparison between RTT samples and WT samples (red line) in iPSC (left panel), neural progenitor cells or NPC (middle panel), and neurons (right panel). The bottom half of each subgraph is the $P$ value from the two-sample $t$ test between MUT/WT and WT/WT. Bins with FDR < 0.05 are shown as a red dot. The red dotted line in the bottom of the subgraphs indicates the minimum $−\log_{10}(P$ value) that corresponds to a FDR (false discovery rate) < 0.05. The blue and red ribbons in correspond to one-half of one standard deviation of each bin for the comparison of WT/WT and MUT/WT, respectively. **b** Microarray analysis of human RTT brain samples compared to age-matched control for frontal cortex[29]. Comparison of gene expression trends vs. gene length in the pooled sample from 2- and 4-year-old patients (left panel; blue line) and the whole dataset (left panel; red line). Observed changes in gene expression vs. gene length in the sample from 5-year-old (right panel; green line) or 8-year-old RTT patient (right panel; purple line). **c** Microarray analysis of RTT human frontal cortex samples[31] compared to controls (left panel) and RTT human temporal cortex samples[31] compared to controls (right panel). **d** RNA-seq analysis from RTT human (female) frontal cortex samples compared to controls (left panel) and RTT human (male) frontal lobe sample compared to controls (right panel). The lines in **a–d** represent the fold-change in expression for genes binned according to gene length (bin size of 200 genes with shift size of 40 genes[6]). Please refer to Table 1 for the total number of samples used for the comparison between two random sets of WT samples and between WT and RTT samples

RNA-seq and microarray datasets is caused by a length-dependent bias in PCR amplification.

**PCA plot confirms reciprocal relationship in MeCP2 datasets.** One of the most intriguing components of the long gene story in RTT is the presence of a reciprocal pattern in the *MECP2-*

overexpression model, where a reported preference for down-regulation of long genes complements the upregulation of long genes reported in *Mecp2*-null mice[6]. To understand this reciprocal relationship, we divided human brain reference samples (B) into three groups ($n = 16$ each) based on different library preparation ID numbers from the Novartis SEQC dataset. The principle component analysis (PCA) plot clearly clustered the

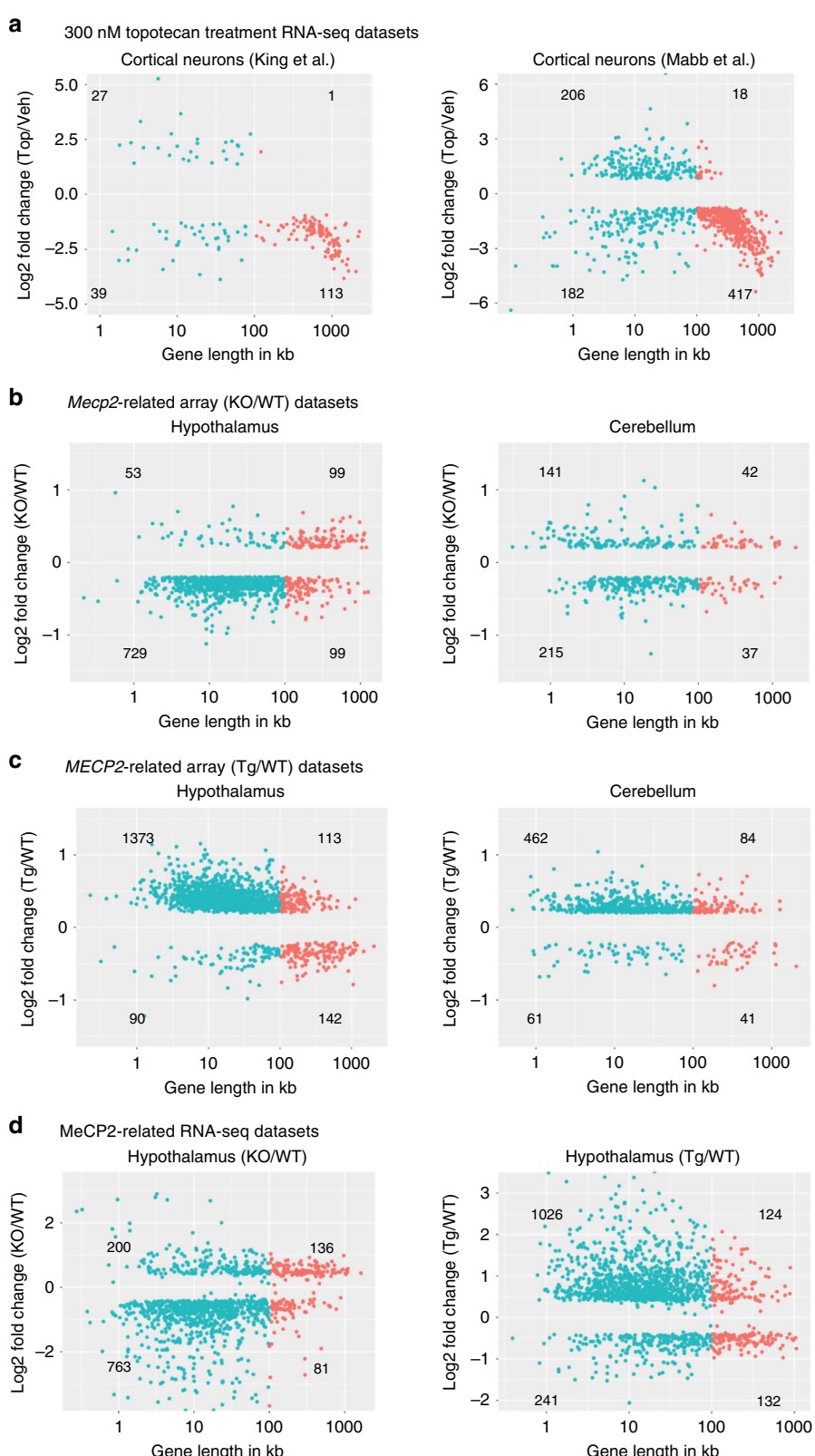

brain samples based on the library preparation group to which they belonged (Supplementary Fig. 9a). Comparing the brain samples of library preparation ID 2 (green) to library preparation ID 1 (red) and ID 3 (blue) separately reversed the running average plot (Supplementary Fig. 9b, c). These results show that a reciprocal relationship can be observed in the gene expression data between any groups that form three distinct clusters on a PCA plot.

We next assessed the influence of the fold-change threshold on differential expression analysis using brain samples (having 750 genes present in Human NanoString nCounter dataset). Although we did not expect to see a trend between replicates, we observed preferential regulation of long genes when the fold change was small (<10%, or log2FC ~13%; Supplementary Fig. 9d). The bias was similar to the trend observed in previously published *Mecp2*-null and overexpression (Tg) models when library preparations ID 2 (red) and ID 1 (green), or library preparations ID 3 (blue) and ID 1 (green), were compared[5,6]. However, we observed no differentially expressed genes when comparing two batches of brain samples from the NanoString nCounter dataset.

In this analysis, all the samples were technical replicates of the same reference RNA and were expected to have identical gene expression levels, but variation associated with library preparation resulted in the samples not clustering together and allowed us to observe an inverse trend in long genes (Supplementary Fig. 9a). Just as biological variation can lead to separation on a PCA plot, so can technical variation, and both can result in the same apparent long gene bias observed in *Mecp2* datasets. Our analysis suggests that differentially expressed genes can be highly variable, with small fold changes, which underscores the importance of proper fold-change cut-offs in differential gene expression analysis.

**Concordance of NanoString and RNA-seq depends on fold change.** To determine whether a long gene trend is present only in the *Mecp2* RNA-seq dataset and not in the NanoString dataset, we generated RNA-seq (>90 million paired-end sequencing reads per sample; $n = 3$ per genotype; Supplementary Data 3) and NanoString ($n = 3$ per genotype; Supplementary Data 4) datasets on cerebellar tissue from WT and *Mecp2*-null mouse models (KO). The PCA plot on normalized datasets showed that the samples clustered based on sample type (Supplementary Fig. 10a, left panel and Supplementary Fig. 10b). Transcriptome analysis was performed using DESeq2[33] on both datasets. We first analyzed RNA-seq data to estimate the strength of the long gene trend. Although there appeared to be a long gene trend in the KO/WT comparison, an overlap plot confirmed there was no significant upregulation of long genes (Supplementary Fig. 10a, middle panel). Consistent with our previous findings, there was no preferential upregulation of long genes in our differential expression analysis (Supplementary Fig. 10a, right panel; absolute $\log_2 FC > \log_2(1.2)$ and FDR < 0.05).

We performed further analysis using a list of 750 genes (~159 long and ~591 short) quantified by both the RNA-seq and nCounter NanoString expression analyses (Supplementary Fig. 10b; Methods). Comparison of the log fold-changes using the classic method (i.e., $\log_2((mean(group1) + 1)/(mean(group2) + 1))$) and using shrunken log fold-changes by DESeq2 (i.e., obtaining reliable variance estimates by pooling information across all the genes) suggested that the latter method yields more highly correlated fold changes (Supplementary Fig. 10c). This is consistent with previous findings showing that shrunken log fold changes are more reproducible[33,37]. Even with this method, however, we observed high variability among genes with low fold-changes between the two datasets, regardless of whether they were long or short (Fig. 5a, b). Moreover, genes with high fold changes in expression (~FC > 20%) were consistently called as differentially expressed in both the datasets (Fig. 5a, b).

This analysis suggests that the genes identified as differentially expressed by RNA-seq with lower fold changes are not reproducible by NanoString. To determine whether fold-changes are inflated in RNA-seq, we compared the absolute difference of $\log_2$ fold change between the RNA-seq and NanoString datasets. We observed fold changes of long genes to be overestimated by RNA-seq technology (Fig. 5c; chi-square test; $P$ value < 7.44e−3), which further supports our hypothesis that artefactual long gene trends are more likely to appear in amplification-based expression datasets.

## Discussion

Several recent papers have suggested that diseases associated with synaptic dysfunction tend to preferentially involve misregulation of long genes (genes with length > 100 kb)[2,5,6,10,19]. To establish a statistical baseline for the length-dependent gene regulation analysis, we took advantage of a large number of SEQC consortium datasets where the relative gene expression fold change has been measured using RNA-seq and microarray. We demonstrated the power of big data analyses by uncovering major sources of technical variation, such as intra-sample variation and PCR amplification bias, that can affect the analysis of long gene expression. By contrast, NanoString nCounter technology, which does not rely on amplification, revealed no long gene bias. Our results demonstrate that amplification-based transcriptomic technologies can lead to overestimations of long gene expression changes.

This is not to say that there is never a bias toward expression changes in long genes. The topotecan datasets showed an authentic long gene trend even after accounting for baseline variability. This sizeable effect on long gene expression is consistent with the biological function of topotecan, which inhibits topoisomerase I; long genes should, in theory, be more dependent on proper unwinding during transcription elongation[10]. By contrast, we found no bias toward long gene dysregulation in the MeCP2 datasets after baseline correction, even when we focused

**Fig. 3** Differentially expressed genes show length-dependent misregulation in topotecan datasets but not in MeCP2 studies. **a** Scatter plot of log fold change in expression between topotecan and vehicle-treated cultured cortical neurons (*y*-axis) against its gene length (*x*-axis) in RNA-seq data from King et al.[10] (left panel; $n = 5$ each; FDR < 0.05) and Mabb et al.[19] (right panel; $n = 3$ each; FDR < 0.01). **b** Scatter plot of log fold change in expression (microarray) between C57BL KO and its C57BL WT littermates (*y*-axis) against its gene length (*x*-axis) in hypothalamus[22] (left panel; $n = 4$ each; FDR < 0.05 and $\log_2 FC > 0.2$) and cerebellum[21] (right panel; $n = 4$ each; FDR < 0.05 and $\log_2 FC > 0.2$). **c** Scatter plot of log fold change in expression (microarray) between FVB Tg to its FVB WT littermates (*y*-axis) against gene length (*x*-axis) in hypothalamus[22] ($n = 4$ each; FDR < 0.05 and $\log_2 FC > 0.2$) and cerebellum[21] ($n = 4$ each; FDR < 0.05 and $\log_2 FC > 0.2$). **d** Scatter plot of log fold change in expression between KO or Tg and WT littermates (*y*-axis) against gene length (*x*-axis) in RNA-seq datasets: Hypothalamus KO/WT comparison[23] (left panel; $n = 3$ each; FDR < 1e−5) and hypothalamus Tg/WT comparison[23] (right panel; $n = 3$ each; FDR < 1e−5). Red dots indicate long genes and blue dots indicate short genes. Differentially expressed genes were obtained from the published gene lists. See Supplementary Fig. 5 for additional analyses from published MeCP2 datasets

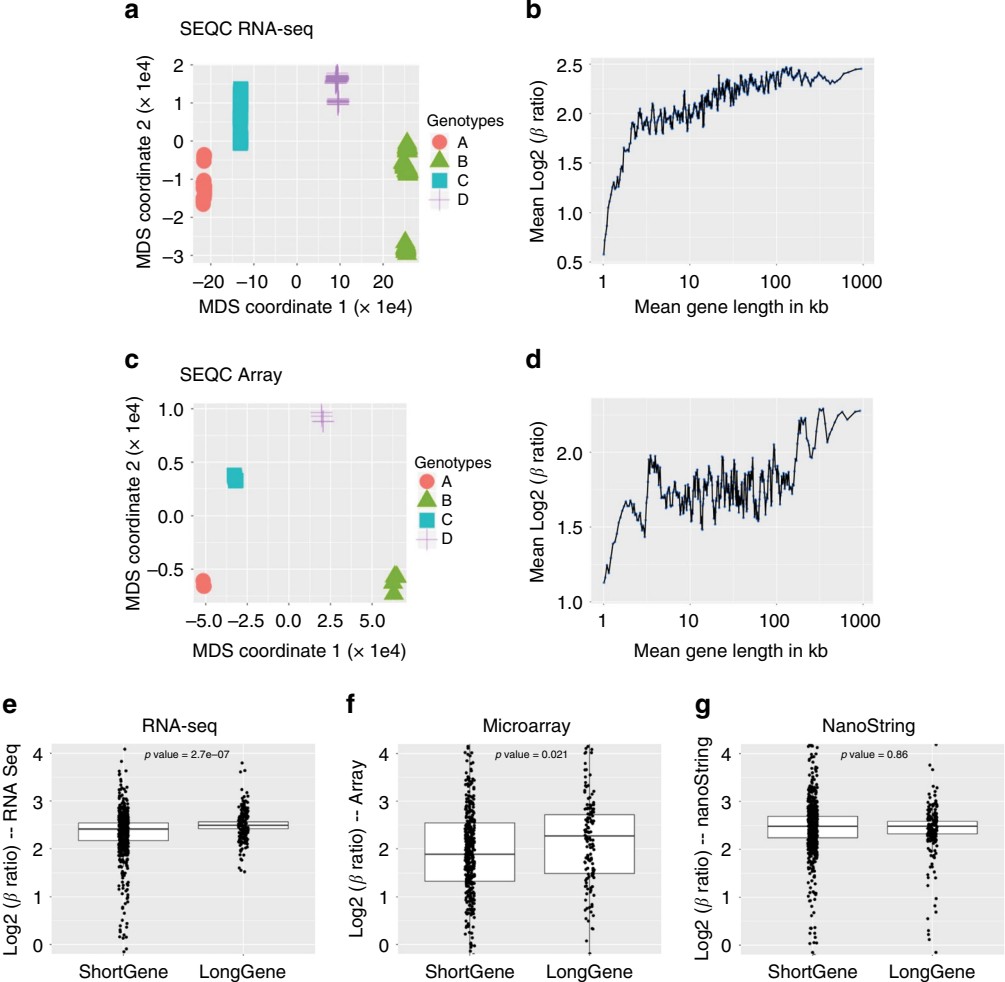

**Fig. 4** A long gene bias is observed in SEQC RNA-Seq and microarray, but not in NanoString, datasets. **a** MDS plot using Euclidean distance on the Novartis SEQC[32] count dataset (A is Universal Human Reference RNA, B is Human Brain Reference RNA, C is a mixture of A and B at a ratio of 3:1, and D is a mixture of A and B at a ratio of 1:3). **b** Mean Log$_2$ $\beta$ ratio of gene expression in the SEQC RNA-seq dataset was plotted against gene length ($\beta$ ratio = (B–A)/(C–A); $n = 64$ each). Each blue dot is a bin of 200 genes with shift size of 40 genes[6]. **c** MDS plot using Euclidean distance on the SEQC microarray dataset. **d** Mean Log$_2$ $\beta$ ratio of gene expression in the SEQC microarray dataset was plotted against gene length ($n = 4$ each). Each blue dot is a bin of 200 genes with shift size of 40 genes[6]. **e–g** Box plots of the ~680 genes that are common amongst the three different platforms. The distributions of the mean Log$_2$ $\beta$ ratio was calculated for each of the ~680 genes in the samples and plotted for long and short genes are compared across the three platforms: (**e**) RNA-seq, (**f**) microarray, and (**g**): NanoString ($n = 6$ each). $P$ values were computed using the Wilcoxon–Mann–Whitney test. In the box plots, the center lines show medians. Box limits indicate the 25th and 75th percentiles, and whiskers extend to 1.5 times the interquartile range. See Supplementary Figs. 6–9 for additional analyses of the SEQC and NanoString datasets

on only those genes that are differentially expressed to a statistically significant degree. The sole exception was the one infantile RTT case, but a single case does not allow us to draw any firm conclusions. Again, this does not rule out that MeCP2 regulates some long genes; it simply does not support a preferential misregulation of long genes in MeCP2-mutant samples.

Apparent expression changes in long genes are clearly liable to exaggeration by biases in microarray and RNA-seq. We recommend eliminating confounds such as batch effects and properly estimating both intersample and intrasample variations; the control datasets must be carefully analyzed in order to reveal the degree of baseline variability, which then can inform further analyses of the size of the signal required to overcome background noise in sequencing datasets (Supplementary Fig. 1). These findings are applicable to all research that utilizes current microarray and sequencing technologies. We hope that revealing the influence of protocols and technologies on RNA-sequencing data will lead to improved technologies and more reliable analyses for amplification-based sequencing data.

## Methods

**Mice**. All mice used in this study (Fig. 5 and Supplementary Fig. 10) were FVB.129 F1-hybrids. They were group-housed with up to five mice per cage. They were maintained on a 14 h light:10 h dark cycle (light on at 06:00) with standard mouse chow and water *ad libitum* in our AAALAS-accredited facility. All research and animal care procedures were approved by the Baylor College of Medicine Institutional Animal Care and Use Committee.

**Postmortem human RTT samples**. Human postmortem samples used in this study (Fig. 2d) were obtained from the NIH BioBrainBank by William Lowry's lab at University of California, Los Angeles (UCLA).

**Analysis of MeCP2 datasets**. The transcriptome datasets from *Mecp2* studies generated using microarray technology (GEO accession IDs: GSE50225, GSE11150, GSE15574, GSE33457, GSE42895, GSE42987, GSE8720, and GSE6955) were downloaded from GEO. RMA function[38,39] in the R *affy* package was used to perform background correction, normalization, and summarization of core probesets. NetAffx annotation files (Release 33 for mm9) was used to map affy probes to their official gene symbols. The expression values for genes with multiple probes were obtained by taking the average log$_2$ expression value across all the probes corresponding to each gene. The NetAffx annotation file has information about the probe location, length and gene coordinates; we calculated gene length using the

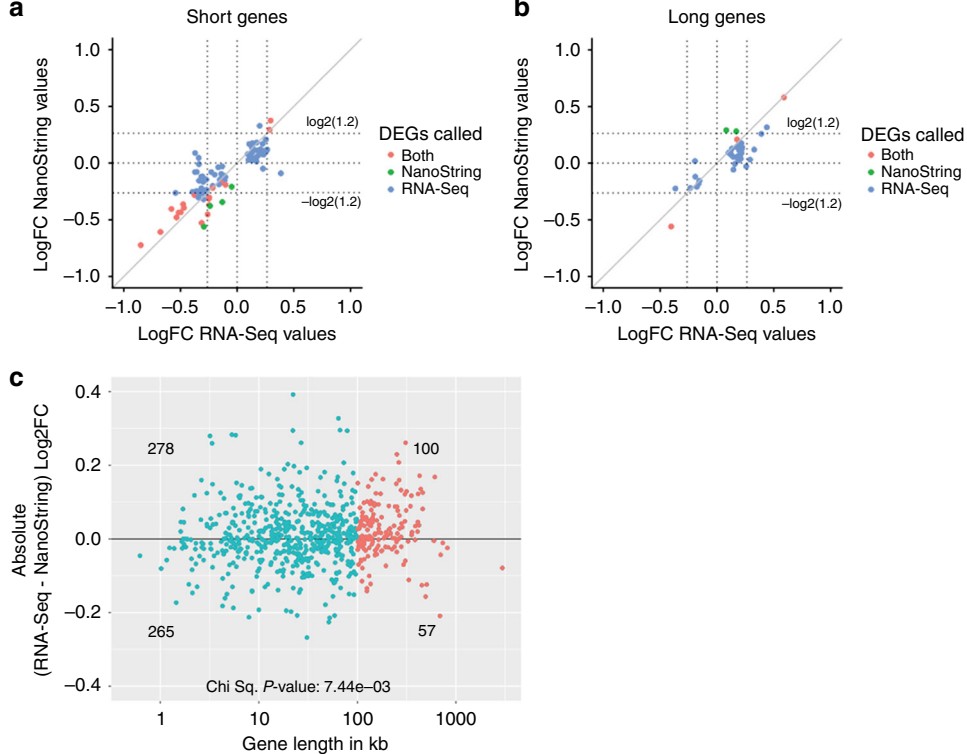

**Fig. 5** Expression changes are overestimated in RNA-seq datasets. Comparison of log fold-change in expression between RNA-seq ($n = 3$ each) and NanoString ($n = 3$ each) for short genes (**a**) and long genes (**b**). Genes were considered to be differentially expressed if FDR < 0.05. Red dots indicate genes found to be differentially expressed in both platforms. The green and blue dots indicate genes that are only found to be differentially expressed in either the NanoString or the RNA-seq dataset, respectively. The dashed lines in (**a**, **b**) on x-axis and y-axis correspond to $\log_2$ fold change of $-\log_2(1.2)$, $\log_2(1)$, and $\log_2(1.2)$. **c** Absolute log fold-change difference between RNA-seq and NanoString (y-axis) plotted against gene length (x-axis). Red dots are long genes, and blue dots are short genes. P values were computed using a chi-square test. See Supplementary Fig. 10 for additional analyses related to this figure

gene coordinates, and we specifically used gene length in all our figures where Gene length in kb is defined on the x-axis. We also ran our analysis on the transcript length (see Supplementary Fig. 6a, b, right panel, and Supplementary Fig. 7a, b, right panel). The extent of length-dependent bias with transcript length was similar to that of gene length. Since gene length information was not available for the Affymetrix Human Genome U95 version 2 array, we mapped the probes to their respective genes and gene lengths using Ensembl Biomart database (version GRCh38.p5/Ensembl Genes 84).

The transcriptome dataset of the visual cortex[6] (GSE67294), RTT iPSC[28] and RNA-seq RTT dataset[28] were mapped to mm10 genome using STAR aligner v2.4.2a[40] (https://github.com/alexdobin/STAR/releases/tag/STAR_2.4.2a). For the hypothalamus RNA-seq dataset[23] (GSE66871), we used a published list of differentially expressed genes and normalized counts. For the Johnson et al.[8] RNA-seq dataset (GSE83474), we used the raw count files provided by the authors in GEO and it was normalized using DESeq2[33] (https://bioconductor.org/packages/release/bioc/html/DESeq2.html). Similarly, for the transcriptome analysis of frontal and temporal cortex from RTT patients, we used the normalized gene expression profile provided in GSE75303[31]. We generated box plots and MDS plots to check for outliers in the sample distributions. The annotation files provided by GPL10558 were obtained to map Illumina probes to official gene symbols and RefSeq hg19 annotation was used to obtain gene length information.

**Running average plots**. We used the same method as described in ref. [6] to compute the running average plots. In brief, the genes were sorted by their lengths and partitioned into bins using a sliding window of 200 consecutive genes in steps of 40 genes. The $\log_2$ fold-change values for genes within each bin were averaged. For consistency with the previous studies, we used genes whose lengths are between 1 and 1000 kb for all the plots. These plots were created using the ggplot2 package (https://github.com/tidyverse/ggplot2) in R.

**Confidence interval estimation in overlap plots**. We define the plots used in Fig. 1 as overlap plots, meaning an overlap of two running average plots that shows intrasample variation between control samples (WT) and intersample variation between two genotypes or conditions. To determine the amount of intra-sample

variation, we computed the standard deviation of the genes in the same sliding window. By definition, the 95% confidence interval for the mean is equal to the sample mean plus or minus 1.96 times standard deviation. In the vast majority of the overlap plots for the MeCP2 datasets, however, the confidence interval of KO/WT (or Tg/WT or MUT/WT) and WT/WT completely overlap. For the sake of legibility, we plotted only sample mean plus or minus half of one standard deviation for each bin in the comparison of WT/WT (or V/V; shown by the blue ribbons) and KO/WT (or Tg/WT or D/V or MUT/WT; shown by the red ribbons). Two-sample Student $t$ test was applied to each of the bins between KO/WT (or Tg/WT or D/V or RTT/WT) and WT/WT (or V/V), followed by multiple hypothesis adjustment using the Benjamini–Hochberg method (FDR). The significant bins (FDR < 0.05) are denoted by red and non-significant bins are denoted by gray. The overlap plots were created using the cowplot package in R (https://github.com/wilkelab/cowplot).

**Gene length distribution in MeCP2 datasets**. To measure the distribution of the long gene bias among differentially expressed genes, we extracted published lists of genes found to be significantly activated or repressed by MeCP2 across different brain regions. The published lists of differentially expressed genes were downloaded from the supplementary files in each study. Because of the frequent changes in gene name and annotation, we used MGI batch query[41] to facilitate uniform comparison between these gene lists. The genomic locations were obtained for mm10/GRCm38. The original fold change and FDR thresholds reported by respective publications were used. For the microarray datasets, genes were plotted against their length. In the case of the RNA-seq datasets, the calculation was done based on UCSC transcript IDs. Long genes (gene length > 100 kb) were represented as red dots and short genes were shown as blue dots. The numbers of the upregulated and downregulated long/short genes are shown in each respective quadrant.

**Analysis of SEQC datasets**. We measured the long gene fold-change bias in RNA-seq and microarray benchmark SEQC consortium datasets, using the RNA-seq datasets generated by all of the Illumina HiSeq 2000 sites and the microarray datasets generated by USF using the Affymetrix Human Gene 2.0 ST Array. The

RNA-seq raw count files and microarray PrimeView normalized files were accessed from the Gene Expression Omnibus database (GEO)[42]. The GEO accession IDs for the RNA-seq and microarray datasets are GSE47774 and GSE56457, respectively. Raw count files from the Australian Genome Research Facility (AGR), Beijing Genomics Institute (BGI), Weill Cornell Medical College (CNL), City of Hope (COH), Mayo Clinic (MAY), and Novartis (NVS) were normalized using the DESeq2 method[33]. Principal Component Analysis (PCA) and MDS plots (using Euclidean distance) were used to do a sanity check for a nominal amount of batch effects.

For further downstream analyses, we decided to use the Novartis dataset, as it had a minimal amount of non-biological variation. The Novartis dataset consisted of 64 technical samples each of A (Universal Human Reference RNA), B (Human Brain Reference RNA), C (3A:1B) and D (1A:3B). We did not use sample type E (Ambion ERCC Spike-In Control Mix 1) or F (Ambion ERCC Spike-In Control Mix 2) in our analysis. For consistency with the SEQC consortium, we used hg19 iGenome NCBI/RefSeq annotation (build 27.2). The *transcripts* and *exon* functions in GenomicFeatures Bioconductor package[43] (https://bioconductor.org/packages/release/bioc/html/GenomicFeatures.html) were used to obtain the gene and transcript lengths, respectively, from the hg19 GTF file. Since a small number of genes or transcripts have multiple different genomic locations, genes or transcripts with the longest length were used. Expression values for genes with multiple transcript clusters were averaged across all transcript clusters corresponding to each gene. Similarly, for the microarray USF PrimeView dataset, sanity checks were performed using boxplot and MDS plots. Boxplots were used to check if the dataset was properly normalized and MDS plots were used to confirm that the dataset had a nominal amount of batch effects or non-biological variation.

**Normalization using total count and TMM methods**. To ensure that our normalization methods were not obscuring a genuine long gene bias, we normalized the raw counts from the Novartis RNA-seq dataset using two other methods apart from DESeq2[33]: 1) Total Counts[44] and 2) the trimmed mean of *M*-values (TMM) method implemented in edgeR[36,37] (https://bioconductor.org/packages/release/bioc/html/edgeR.html). For Total Counts, scaling factors were computed such that the normalized read counts across all samples are equal. In the case of the TMM method, we used the *calcNormFactors* function in the edgeR Bioconductor package to get the scaling factors and normalized read counts.

**SEQC NanoString sample preparation and analysis**. We purchased Universal Human Reference RNA from Agilent Technologies, Inc., and Human Brain Reference RNA from Life Technologies, Inc. For the nCounter experiments, we used the same RNA sample types as the SEQC consortium. We assessed RNA purity and integrity with Bioanalyzer (Agilent Technologies, Inc.) prior to use in the nCounter assays. Sample preparation and analysis were done using a nCounter Prep Station 5 s and a nCounter Digital Analyzer 5s. Expression of 770 genes (~730 genes with ~40 housekeeping genes and positive and negative controls) was assessed using the nCounter Human PanCancer Pathways Panel. A second Pan-Cancer Pathways Panel was run using the same samples submitted to the first panel to assess the effect of batches on nCounter results. We used the NanoStringNorm function[45] in the R NanoStringNorm package (https://cran.r-project.org/web/packages/NanoStringNorm/index.html) to normalize the dataset. Boxplots and MDS plots were used to confirm the samples were properly normalized and to check for the presence of batch effects. The two-sided Wilcoxon rank sum test was used to compare the distribution of the fold-change between long and short genes across the three different platforms—RNA-seq, Microarray and NanoString.

**Cerebellar gene expression from *Mecp2*-null and WT mice**. We performed RNA extraction and purification from the cerebellum of male mice 8–9 weeks of age (three biological replicates from WT mice and three biological replicates from *Mecp2*-null mice) using the Aurum™ Total RNA Fatty and Fibrous Tissue Kit (Bio-Rad 7326830) per the manufacturer's instructions. Genomic DNA was eliminated using an on-column DNase digestion step. RNA quality was assessed using the Agilent 2100 Bioanalyzer system prior to library preparation for deep sequencing or use of the total RNA for NanoString nCounter quantification.

RNA sequencing was performed using Illumina HiSeq 2000. All sequencing was done by the Genomic and RNA Profiling Core at Baylor College of Medicine. For each sample, about 90–110 million pairs of 100 bp reads were generated. Raw reads were aligned to the *Mus musculus* genome (Gencode mm10; version M10) using STAR aligner v2.4.2a[40] with default parameters. The overall mappability for all 6 samples was above 90% (Supplementary Data 3). The read counts per gene were obtained using the *quantMode* function in STAR. These read counts are analogous to the expression level of the gene. Using the obtained raw counts, normalization and differential gene analysis were carried out using the DESeq2 package in the R environment. DESeq2 allows us to test for gene expression changes between samples in different conditions using more robust shrinkage estimation for dispersion and fold changes[33]. The default negative binomial generalized linear model with Wald test implemented in the package was used to identify significant differentially expressed genes. Log fold change was calculated using both the classic method and shrinkage estimates calculated by DESeq2.

For the nCounter experiments, sample preparation and quality analysis were done using a nCounter Prep Station 5s and an nCounter Digital Analyzer 5s. Expression of 784 genes (750 endogenous genes with 34 housekeeping genes and positive and negative controls) was assessed using the nCounter Mouse PanCancer Pathways Panel (Supplementary Data 4). We used NanoStringNorm function[45] in the R NanoStringNorm package to normalize the dataset and DESeq2 for differential expression analysis.

**Code availability**. The custom R scripts to reproduce all the results/plots used in this manuscript are available at the following webpage: https://doi.org/10.5281/zenodo.1226607.

**Data availability**. RNA-seq and NanoString datasets that support the findings of this study have been deposited in the Gene Expression Omnibus (GEO) database with the accession codes as follows: GSE94073 [https://www.ncbi.nlm.nih.gov/geo/query/acc.cgi?acc=GSE94073], GSE105047 [https://www.ncbi.nlm.nih.gov/geo/query/acc.cgi?acc=GSE105047], and GSE107399 [https://www.ncbi.nlm.nih.gov/geo/query/acc.cgi?acc=GSE107399]. GEO accession codes and details of all the public datasets used in this manuscript are available in Supplementary Data 1 table.

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

## Acknowledgments

We thank Laura Lavery, Rami Al-Ouran, Laura Lombardi, Ezequiel Sztainberg, Aya Ishida, and Vicky Brandt for helpful discussions and suggestions. We also wish to thank the Quantiative & Computational Biosciences and the Medical Scientist Training Program at Baylor College of Medicine. This project was supported by the Genomic and RNA Profiling Core at Baylor College of Medicine and the expert assistance of the Core Director, Lisa D. White, Ph.D. This project was funded by the National Institutes of Health (5R01NS057819 to H.Y.Z., 5R01GM120033 and 1R01AG057339 to Z.L.), the Howard Hughes Medical Institute (H.Y.Z.), March of Dimes (W.E.L.), the Frontiers Group of the Paul Allen Family Foundation (W.E.L.), Robert and Janice McNair Foundation (A.E.P.), Baylor Research Advocates for Student Scientists(A.E.P.), Cancer Prevention Research Institute of Texas (RP170387 to Z.L.), Chao Family Foundation (Z.L.), and Huffington Foundation (Z.L.).

## Author contributions

Study concept and design: Z.L., A.T.R., and H.Y.Z.; methodology and investigation: A.T.R. and Z.L.; data curation: A.T.R.; computational analysis: A.T.R.; interpretation of data: A.T.R., Z.L., Y.W.W., and H.K.Y.; visualization: A.T.R. and Y.W.W.; acquisition of new data for validation: A.T.R., W.E.L., and A.E.P.; drafting of the manuscript: A.T.R., Z.L., and A.E.P.; critical revision of the manuscript: Z.L., H.Y.Z., A.T.R., Y.W.W., and A.E.P.; supervision: Z.L. and H.Y.Z.; funding acquisition: Z.L. and H.Y.Z. All the figures were generated by A.T.R.

## Additional information

**Competing interests:** H.Y.Z. is one of the co-holders of U.S. Patent 6,709,817 (Method of Screening Rett Syndrome by Detecting a Mutation in MECP2, March 23, 2004). All other authors declare no competing interests.

