## [Peer Review File · Nature Communications]

Reviewers' comments:

Reviewer #1 (Remarks to the Author):

This is a relatively technical study that shows that gene expression profiling using standard methods can lead to a bias towards differential expression of long genes due to the amplification of the samples. The authors suggest to use randomized control samples to correct for this bias.

General comment:

The paper has a clear message but it is presented by multiple methods with no clear motivation for each method. Across the paper, the authors show that there is a bias in differential expression of long genes but they show it with a different methods and presentations that change along the paper - It is not clear why?

Specific comments:

1. The method relies on dividing the genes to arbitrary bins. Are there another way to analyse the data as continuous quantitative trend without bins?
2. Not clear to me why the methods and presentation in Figure 2 are different from Figure 1?
3. Not clear to me why the authors decided to repeat the analysis in a different way and to present it in a new section ("Differential gene expression analysis for Topotecan and Mecp2 datasets", Figure 3). It is similar argument and basically shows the same thing again. My suggestion is to combine it with the sections before ("Baseline length dependency should first be estimated from the control groups: Topotecan studies as positive controls", "Gene length trends do not hold up in datasets for MeCP2 mouse models")
4. Not clear to me why in Figure 4 the authors choose to use a new measure – β to show the effect of gene length.

Reviewer #2 (Remarks to the Author):

The authors of the Raman et al manuscript argue that the numerous reports about preferential dysregulation of long genes associated with neurological diseases are largely based on failures to account for gene-length related biases in the techniques for measuring gene expression.

The presented is argument is really well constructed and thorough. Large number of published datasets are re-examined in new data was generated when needed. A statistical approach is presented that uses the case-control vs control-control design to properly test for gene-length-dependent mis-regulation. The statistical approach was validated on a Topotecan studies where the long-gene downregulation of samples with deficient topotecan was confirmed. When applied to numerous data sets related to Rett syndrome the technique showed no convincing length depended mis-regulation, contrary to the original reports.

Technically the manuscript is strong and the text and figures of high quality. I have nothing of substance to suggest.

The manuscript attempts to correct the record on an important topic and I am convinced it should be published. I am curious to see if counter arguments would follow.

We appreciate the reviewers' reading of the manuscript. Below, for the sake of clarity, we quote the reviewers' original comments in italics, and respond in regular font.

Reviewer #1

This is a relatively technical study that shows that gene expression profiling using standard methods can lead to a bias towards differential expression of long genes due to the amplification of the samples. The authors suggest to use randomized control samples to correct for this bias.

The paper has a clear message but it is presented by multiple methods with no clear motivation for each method. Across the paper, the authors show that there is a bias in differential expression of long genes but they show it with a different methods and presentations that change along the paper - It is not clear why?

We are glad that the reviewer found the message of the paper clear. The answer to why we used multiple methods on multiple datasets is basically that any study that ends up showing that previous publications were mistaken must be done with extreme thoroughness and care. This is particularly the case here, because the results of the previous papers were so striking (a bias toward misregulation of genes of a certain length) that they became very well-known in the field. We felt that in order to show that this was not the case, it would be necessary to analyze the different sets of data in multiple ways to establish that we are being as even-handed and scrupulous as possible.

We have combed through the manuscript to revise it for greater clarity. It should be easier now to follow the rationale for each set of experiments.

Specific comments:

1. The method relies on dividing the genes to arbitrary bins. Are there another way to analyse the data as continuous quantitative trend without bins?

Analyzing the data in bins or windows is the standard methodology for analyzing genomic and epigenomic data (as established originally by King and colleagues⁴ and in the original analyses of the MeCP2 datasets¹⁻³). The reason is that if you were to try to analyze the data on a continuous line, you would be faced with many additional choices: the number of knots in the B-spline, the order of the polynomial, etc. (and some of these choices will be window-based as well). In fact, we used several different window sizes to ensure that no particular bin size was introducing an inadvertent bias. (This is another reason for going through so many analyses of the data.) We also used the very same window sizes as used in the previous publications, to ensure that we were replicating the conditions of the original studies: and we did indeed see the same trend that appeared in those studies (but of course that trend disappears when we do proper statistical analysis to establish the baseline).

2. Not clear to me why the methods and presentation in Figure 2 are different from Figure 1?

They are not different. The only difference is between Fig 2A and 2B&D: for the latter two panels we did not calculate the intra-sample variation and two-sample Student t-test because the number of replicates (or n) is equal to 1 in these human post-mortem studies.

3. *Not clear to me why the authors decided to repeat the analysis in a different way and to present it in a new section (“Differential gene expression analysis for Topotecan and Mecp2 datasets”, Figure 3). It is similar argument and basically shows the same thing again. My suggestion is to combine it with the sections before (“Baseline length dependency should first be estimated from the control groups: Topotecan studies as positive controls”, “Gene length trends do not hold up in datasets for MeCP2 mouse models”)*

The analysis in Figures 1 and 2 relied on arbitrary binning. In Figure 3, we analyzed the data at the level of the individual genes by focusing on those differentially expressed under conditions of MeCP2 perturbations. By taking a different approach on the same dataset, we reproduced our observation and conclude with even more statistical confidence that the apparent "long gene effect" in MeCP2 studies is an artifact.

4. *Not clear to me why in Figure 4 the authors choose to use a new measure – β to show the effect of gene length.*

We used SEQC dataset as a benchmark dataset to assess if the long gene trend is unique to MeCP2 datasets or appears in other RNA-seq datasets. The dataset consisted of four different types of RNA samples: A (Universal Human Reference RNA), B (Human Brain Reference RNA), C (a mixture of A and B at a ratio of 3:1), and D (a mixture of A and B at a ratio of 1:3). We computed β ratio because, in theory, it should be independent of gene length. Since the ratio is 4:1, we expected the average $\log_2 \beta$ ratio to be a horizontal line along the x-axis with a y-intercept equal to two (i.e., $y=2$ on an xy plane) (lines 223-238).

Based on our analysis, we showed that the expected ratio was not maintained for long genes and was overestimated. This indicated that the apparent "long gene trend" is not unique to MeCP2 datasets (Figures 4B and 4D). In other words, not only did the previous studies fail to establish a proper statistical baseline, but they failed to test the hypothesis that other genetic mutations might also produce an apparent bias toward misregulation of long genes.

Reviewer # 2

The authors of the Raman et al manuscript argue that the numerous reports about preferential dysregulation of long genes associated with neurological diseases are largely based on failures to account for gene-length related biases in the techniques for measuring gene expression.

The presented argument is really well constructed and thorough. Large number of published datasets are re-examined in new data was generated when needed. A statistical approach is presented that uses the case-control vs control-control design to properly test for gene-length-dependent mis-regulation. The statistical approach was validated on a Topotecan studies where the long-gene downregulation of samples with deficient topotecan was confirmed. When applied to numerous data sets related to Rett syndrome the technique showed no convincing length depended mis-regulation, contrary to the original reports.

Technically the manuscript is strong and the text and figures of high quality. I have nothing of substance to suggest. The manuscript attempts to correct the record on an important topic and I am convinced it should be published. I am curious to see if counter arguments would follow.

We greatly appreciate the assessment of our study by Reviewer #2, as well as the positive recognition of our statistical approach and the importance of our manuscript to the field.

References:

- 1 Sugino, K. *et al.* Cell-Type-Specific Repression by Methyl-CpG-Binding Protein 2 Is Biased toward Long Genes. *34*, 12877-12883 (2014).
- 2 Gabel, H. W. *et al.* Disruption of DNA-methylation-dependent long gene repression in Rett syndrome. *Nature* *522*, 89-93 (2015).
- 3 Johnson, B. S. *et al.* Biotin tagging of MeCP2 in mice reveals contextual insights into the Rett syndrome transcriptome. *Nat Med*, doi:10.1038/nm.4406 (2017).
- 4 King, I. F. *et al.* Topoisomerases facilitate transcription of long genes linked to autism. *Nature* *501*, 58-62, doi:10.1038/nature12504 (2013).

REVIEWERS' COMMENTS:

Reviewer #1 (Remarks to the Author):

The authors addressed all of my comments.

REVIEWERS' COMMENTS:

We appreciate the reviewers' reading of the manuscript. Below, for the sake of clarity, we quote the reviewers' original comments in italics, and respond in regular font.

Reviewer #1

The authors addressed all of my comments

We are pleased to see that Reviewer #1 feels that our responses have addressed all the comments.